# Predictors of the Occupational Burnout of Healthcare Workers in Poland during the COVID-19 Pandemic: A Cross-Sectional Study

**DOI:** 10.3390/ijerph19063634

**Published:** 2022-03-18

**Authors:** Katarzyna Szwamel, Antonina Kaczorowska, Ewelina Lepsy, Agata Mroczek, Magdalena Golachowska, Ewa Mazur, Mariusz Panczyk

**Affiliations:** 1Institute of Health Sciences, University of Opole, Katowicka 68 Street, 45-060 Opole, Poland; antonina.kaczorowska@uni.opole.pl (A.K.); ewelina.lepsy@uni.opole.pl (E.L.); agata.mroczek@uni.opole.pl (A.M.); magdalena.golachowska@uni.opole.pl (M.G.); 2Faculty of Medicine, University of Technology in Katowice, 40-555 Katowice, Poland; ewam_40@interia.pl; 3Department of Education and Research in Health Sciences, Faculty of Health Science, Medical University of Warsaw, Litewska 14/16, 00-581 Warsaw, Poland; mariusz.panczyk@wum.edu.pl

**Keywords:** psychological burnout, occupational stress, depression, quality of life, health personnel, COVID-19

## Abstract

The study aims at analysing the occupational burnout phenomenon, the level of anxiety and depression, as well as the quality of life (QOL) of healthcare workers (HCW) during the COVID-19 pandemic. There were 497 healthcare workers examined across Poland. The Maslach Burnout Inventory (MBI), Hospital Anxiety Depression Scale (HADS) and World Health Organization Quality of Life Instrument Short Form (WHOQOL BREF) were used. A total of 71.63% (356) of the respondents presented high and moderate levels of emotional exhaustion during the pandemic, 71.43% (355) reported low and moderate job satisfaction levels, whereas 40.85% (203) displayed high and moderate levels of depersonalization. A group of 62.57% (*n* = 311) demonstrated clear or borderline states of anxiety disorders, while 38.83% (*n* = 193) suffered from depression or its borderline symptoms. Direct predictors of occupational burnout included: escalating depression; quality of life domains such as the physical, psychological and social; being provided personal protective equipment (PPE) by an employer; age; medical profession; and material status. Emotional exhaustion appeared to be much higher in nursing and ‘other’ medical professionals than in physiotherapists (*p* = 0.023). In the times of pandemic, the occupational burnout prophylaxis ought to be focused on early recognition of depression like symptoms and their treatment, as well as providing the staff with PPE and satisfying earnings.

## 1. Introduction

Human resources constitute a key element of the safe functioning of healthcare institutions [1]. In the times of the COVID-19 pandemic, working in the healthcare sector has become a source of stress for a large number of workforce, especially physicians, nurses and paramedics who work ‘at the frontline’ [2,3,4,5]. Many aspects of this pandemic have caused moral distress, and unexpected challenges to the ethical values of health professionals, including complex human rights issues in many settings. According to the Centres for Diseases Control and Prevention, possible symptoms that frontline health care professionals may experience in a pandemic include: irritation, anger, lack of motivation, feeling helpless or powerless, feeling sad, depressed or overwhelmed or burned out, having difficulty in sleeping or concentrating, and feeling tired [6]. Negative mental outcomes have, additionally, been aggravated by the media who informed about the pandemic and focused on death rates among healthcare workers and the disease spread in healthcare institutions [4]. Such massive exposure to all kinds of information about COVID-19 implies a possibility of a massive traumatic incident with an unprecedented influence on mental health [7]. These problems we mentioned above affect functioning in the workplace [8]. A body’s reaction to permanent stress, which has its origins at work, results in occupational burnout. Working under pressure might increase emotional exhaustion which, consequently, triggers a defence mechanism called depersonalization (lower sensitivity to others) [9]. Occupational burnout results in higher psychoactive substance intake, depression and an increasing number of suicides [10].

There are numerous definitions of burnout. According to WHO (World Health Organization), ‘burnout is a syndrome conceptualized as resulting from chronic workplace stress that has not been successfully managed. It is characterised by three dimensions: (1) feelings of energy depletion or exhaustion; (2) increased mental distance from one’s job, or feelings of negativism or cynicism related to one’s job; and (3) a sense of ineffectiveness and lack of accomplishment’ [11]. Schaufelli and Enzmann claim that it is a ‘permanent, negative state of work found in healthy professionals and characterised by fatigue accompanied by mental and physical discomfort, the feeling of lower efficiency, decreased motivation or dysfunctional attitudes and behaviours at work’ [12] (pp. 19–41). However, the article is based mainly on the conception stated by Maslach and Leiter. According to them, ‘burnout is a psychological syndrome emerging as a prolonged response to chronic interpersonal stressors on the job’ [13] (pp. 103–111). The authors also indicated three key dimensions of this reaction, such as: an overwhelming exhaustion, feelings of cynicism and detachment from the job, and a sense of ineffectiveness and lack of accomplishment [13].

The relationship between depression and burnout requires a separate discussion. The researchers’ opinions are ambiguous on this point. Due to common etiological pathways and shared symptoms, the singularity of the burnout phenomenon vis-à-vis depression is unclear [14]. Bianchi et al. claim that burnout may not be a separate psychological phenomenon but a dimension of depression [15]. In turn, the findings of a metanalysis conducted by Koutsimani et al. revealed no conclusive overlap between burnout and depression and burnout and anxiety, indicating that they are different and robust constructs [16]. In the study of Pachi et al., the association between burnout and depression was confirmed. These authors believed that, despite the fact that a strong correlation coefficient was obtained between these variables, the interpretation of the variation of depression from burnout at the level of 43.7% cannot justify the overlap between burnout and depression. The authors adopted the position of the World Health Organization, which treats burnout as a separate disease [17]. We also adopted this concept in our study.

The reasons for burnout may be sought in three areas: an individual one (age, gender, education, marital status, low mental resistance, low self-esteem, no sense of safety or no satisfaction in personal life), an interpersonal one (worker–client relationship, inability to achieve balance between caring for oneself and caring for others, competition, psychological abuse, mobbing) and an organizational one (work overload, low earnings, lack of PPE and bad working conditions). The terms mentioned above might, but do not have to, facilitate occupational burnout. It all depends mainly on an individual worker’s characteristics and the situation itself [18].

Due to the COVID-19 pandemic, medical workers have been constantly subjected to a number of stressors at their workplace that may significantly affect the areas described above. They include a high death rate of COVID-19 patients, a greater number of overtime shifts, fear of not getting appropriate medical equipment (including PPE), being endangered with COVID-19 at their workplace, a chance of COVID-19 transmission and infection of family members, no access to lab tests in cases of COVID-19 symptoms, fear of the virus spreading at their workplace, insecurity, fear of no access to babysitting while doing overtime or in case of schools’ shutdown, no support in other personal and family aspects in case of increasing work requirements, fear of poor communication, feelings of guilt relating to the lack of contribution, uncertainty about the future of the workplace or employment, learning new technical skills, and adapting to a different workplace or schedule [3,5,19]. Additional medical staff issues comprised troublesome working conditions connected with the need to wear extra protective suits; issues related to meeting physiological needs; temporary relocation away from their families (hotel accommodation specially designed for COVID-19 hospital staff); an inevitability of facing patients, their caretakers and families’ reactions related to the disease, hospitalization; and burdensome contact with family members caused by suspending visitations. Many medical workers have experienced discrimination as well as social and even family rejection. Such behaviours were especially aimed at the medical staff of so called COVID-19 hospitals and contagious disease wards. The situation was, additionally, worsened by the issues that existed in the healthcare system before the pandemic, such as staff shortages, low earnings, system insufficiency and the negligence of key issues during the previous years. All the above negatively affect the implementation of medical staff duties during the pandemic [20].

The research that thoroughly examined the psychological impact of the epidemic such as SARS in 2003 (acute respiratory syndrome) reported that 10% of medical staff suffered the symptoms of high post-traumatic stress disorder (PTSD) three years after the epidemic breakout [21]. Considering the larger range and much higher death rate of COVID-19 patients, the pandemic might affect the mental health of healthcare workers to a much greater extent. The occurrence of a worldwide pandemic such as COVID-19 might be regarded as a traumatic incident [22]. The research conducted in Europe and across the world among medical staff during the COVID-19 pandemic revealed that it causes anxiety disorders and depression, increases the level of occupational burnout and enhances emotional exhaustion and the level of depersonalization as well as decreases job satisfaction [23,24,25,26,27].

Bearing in mind the stressors mentioned above and potential traumatic situations that healthcare workers are subjected to at workplace during the COVID-19 pandemic, we decided to examine its psychological effects. We, additionally, decided to analyse the occupational burnout phenomenon, the level of anxiety and depression, as well as the quality of life among healthcare workers during the COVID-19 pandemic. We acknowledged that it is worth analysing the phenomena not only among nurses and physicians working ‘at the frontline’, but also midwives and physiotherapists who work with COVID-19 patients on daily basis. It seemed interesting to compare the results between the groups of professionals. The results could be exploited in order to plan a strategy aimed at creating a safe work environment in the times of a pandemic, designing procedures focused on reducing negative aspects of pandemic stressors affecting the mental health of healthcare workers with respect to a job specification, as the references available so far have not included midwives or physiotherapists.

The aim of the study were: (1) the analysis of the burnout phenomenon, the level of anxiety and depression and the quality of life among healthcare workers in the times of the COVID-19 pandemic and (2) the establishment of the factors significantly determining the level of occupational burnout in this group.

## 2. Materials and Methods

### 2.1. Study Design and Setting

This was a cross-sectional study conducted from June 2020 to January 2021. The examination was carried out according to the diagnostic survey method across Poland. The research project was specifically approved by the Bioethics Committee of Opole Medical School (no 7/PI/2020). The STROBE guidelines (Strengthening the Reporting of Observational Studies in Epidemiology) were followed.

Health at a Glance data shows that, before the COVID-19 pandemic, countries in Europe, such as Poland, Latvia and Romania, had fewer doctors and nurses per population comparatively to Germany, Norway or Sweden, etc. Therefore, Poland had less capacity to respond to the pandemic. In Poland, the number of practising doctors per 1 000 population equals 2.4 and practising nurses 5.1, respectively (the average for EU countries is 3.8 for practising doctors and 8.2 for practising nurses). Insufficient human resources made it difficult for healthcare professionals in Poland to respond to the sharp increase in demand for care [28]. Such a situation could initiate or aggravate the burnout syndrome in these people. Sagan et al. claims that Poland have been overly reliant on their relatively high bed capacity, but this could not be supported with a sufficient health workforce capacity. During the period of our research, Poland had not developed effective find, test, trace, isolate and support systems over the summer, despite having relaxed most of the transmission protection measures since late spring. This left Poland ill prepared for the rise in the number of COVID-19 infections they have been experiencing [29].

### 2.2. Participants

The sample includes 497 subjects, recruited by nonprobabilistic sampling. Before completing the questionnaire, the participants were informed about the scope and purpose of the study, as well as about the voluntary and anonymous nature of the answers provided. The participants were invited to fill in the questionnaire in electronic form (CAWI—Computer Assisted Web Interview). The respondents were selected according to the ‘snowball sampling’ method and the questionnaire was posted in social media (e.g., on Facebook, in groups for nurses, physiotherapists and midwives, etc.)

The inclusion criteria involved age ≥ 18, consent for the participation in the study, active work in one of the following medical professions during the COVID-19 pandemic: a nurse, physiotherapist, midwife, physician, paramedic, psychologist, pharmacist, laboratory diagnostician, sanitary–epidemiological station worker, radiologist or medical caretaker. The study excluded non-active healthcare workers, healthcare workers doing jobs other than mentioned above and those who did not provide the consent for the participation in the study. The available data show that 234,117 nurses, 28,899 midwives [30], 145,659 doctors [31], 26,495 physiotherapists are employed in Poland [32]. The Central Statistical Office in Poland reports that 11,000 paramedics work in medical rescue teams [33]. With a confidence level of 95%, a margin of error of 5%, *p* = 50%, the minimum study sample was set at 1896 subjects. Initially, 1508 respondents were interested in the study (they opened the electronic questionnaire). A group of 1008 participants started filling in the form but did not complete it. The questionnaire required the respondents to answer all the questions compulsorily otherwise it could not be sent to the database. Therefore, eventually, 500 people (33.3%) completed all the questionnaires properly. Three of them appeared to be medical field students and their questionnaires had to be excluded from the study. Finally, the research was based on 497 well completed questionnaires (maximum error 4%, confidence level of 95%, *p* = 50%).

### 2.3. Variables and Data Collection

The method of diagnostic survey with the use of a questionnaire was used. The tools included three standardised questionnaires and the authors’ self-prepared one. Maslach Burnout Inventory (MBI) was designed in 1981 by Ch. Maslach and S.E. Jackson [34]. The test evaluates three aspects of burnout syndrome: emotional exhaustion, depersonalization and a decreased sense of self-accomplishment. It consists of 22 test questions assessing the frequency of occurrence of the aspects mentioned above on a 0–6 point scale, divided into three subscales to relate to each aspect of burnout alone [35]. The answers are given according to the 7-point scale of frequency where 0 means ‘never’ and 6 ‘every day’. The score is calculated separately for each subscale by adding the points in each aspect: emotional exhaustion—high (>27), moderate (17–26), low (0–16); depersonalization—high (>13), moderate (7–12), low (0–6); lack of accomplishment—high (0–31), moderate (32–38), low (>39). The higher the score at emotional exhaustion and depersonalization scales, the more intense the burnout is, while the lower the score at the sense of accomplishment scale, the higher the indicator of burnout [34,35].

Hospital Anxiety Depression Scale (HADS) is a commonly used scale to assess depression and anxiety. It has been used to evaluate nursing staff in Poland before [36]. It originally contained 7 positions assessing anxiety and 7 positions related to depression states. After the modification, 2 positions for irritation and aggression were added. All in all, the scale consists of 16 closed questions with 4 possible answer options. Each answer is graded from 0–3 points. The score is then calculated for each category. The categories were distinguished individually for the anxiety and depression subscales (0–7—no disorders, 8–10—borderline state, 11–21—disorders present). The study used a Polish translation validated by M. Majkowicz in which the α-Cronbach coefficient for the anxiety subscale amounted for 0.77–0.80 and for depression subscale 0.84–0.85 [37].

Quality of life was assessed with the Polish version of the World Health Organization Quality of Life Instrument Short Form (WHOQOLBREF) within four domains: D1-Physical, D2-Psychological, D3-Social relationships, and D4-Environmental. WHOQOLBREF consists of 26 questions. The examinees grade each aspect at a 5-grade scale (very bad, bad, neutral, good, very good). The scale includes some questions that are separately analysed: Question 1 applies to general individual perception of one’s QOL, Question 2 concerns general individual perception of one’s health condition. The domain scoring reflects individual perceptions of the QOL domains and has a positive direction—the higher the score, the higher the QOL. The overall scoring for each domain is calculated by counting the average of all the positions included in each domain. Internal cohesion of Polish version of WHOQOLBREF (α-Cronbach coefficient) is set at 0.90 [38].

The authors’ self-prepared questionnaire comprised 17 questions concerning sociodemographic data such as the number of workplaces, type of profession, being provided PPE as well as the questions related specifically to the COVID-19 pandemic (getting through the infection or quarantine). Employers providing workers with PPE was evaluated at a 0–5 point scale where 0 means ‘no PPE provided’ and 5 ‘PPE fully provided’.

We conducted a full psychometric analysis of all the tools used: MBI, HADS and WHOQOL BREF. All of them have satisfactory psychometric parameters (reliability and construct validity), in line with the theoretical assumptions underlying their development by the authors of these scales.

### 2.4. Ethical Considerations

The study was approved by the Bioethics Committee of Opole Medical School, Poland (No 7/PI/2020). All participants were informed of the study protocol and provided informed consent to participate. The study protocol was developed in accordance with the Declaration of Helsinki.

### 2.5. Statistical Methods

For some statistical analysis the sample group was subdivided into 4 subgroups of nursing staff, midwives, physiotherapists and ‘other’ medical professions (physicians and paramedics). Chi-squared test (with Yates’ correction for 2 × 2 tables) was used to compare qualitative variables among groups. In case of low values in contingency tables, Fisher’s exact test was used instead. Kruskal–Wallis test (followed by Dunn posthoc test) was used to compare quantitative variables between three groups. Uni- and multivariate linear regressions were used to analyse impact of potential predictors on quantitative variables. Regression parameters with 95% confidence intervals were shown. No variable selection was performed since including all potential predictors yields satisfactory SPV (subjects per variable) ratio of approx. 16. Significance level for all statistical tests was set to 0.05. R 4.0.5 was used for computations.

## 3. Results

### 3.1. Characteristics of the Study Group

The sample group included 497 participants out of whom there were 240 nurses (48.29%), 106 physiotherapists (21.33%), 82 midwives (16.50%) and 69 mostly physicians and paramedics, with some single representatives of psychologists, pharmacists, lab diagnosticians, sanitary–epidemiological station workers, radiologists and medical caretakers. The respondents came from all parts of Poland, however, the biggest groups were from Silesian Voivodship (150; 30%), Opolskie Voivodship (124; 24.8%), Lower Silesian Voivodship (70; 14%), Greater Poland Voivodship (45; 9%), Mazowieckie Voivodship (26; 5.2%), Malopolskie Voivodship (15; 3%) and Podkarpackie Voivodship (15; 3%). The average age of the respondents was 40.06 ± 10.62 and they were mostly female (442; 88.93%) and highly educated (279; 56.14%). Most of them resided in cities (372; 74.85%), declared to be in permanent relationships (389; 78.27%) and of a good (248; 49.90%) or average (159; 31.99%) material status. The majority of them were also employed in hospitals other than COVID-19 or contagious diseases hospitals (289; 58.15%). The group of 127 (25.55%) participants were in quarantine during the pandemic and 55 (11.07%) were infected by SARS-CoV-2 virus (Table 1).

### 3.2. Occupational Burnout, Quality of Life and the Level of Anxiety and Depression in Healthcare Workers during the Pandemic

Of all the respondents, 217 (43.66%) displayed a high, 141 (28.37%) low and 139 (27.97%) moderate level of emotional exhaustion. A low level of depersonalisation was found in 294 (59.15%) of the examinees, while moderate in 166 (23.34%) and high in 87 (17.51%) of them. A total of 205 (41.25%) respondents reported a low, 150 (30.18%) moderate and 142 (28.57%) high level of job satisfaction. The average score for QOL was 3.67 points (SD = 0.79) and for health condition perception 3.54 points (SD = 0.83). The highest scores were found in the psychological domain (14.46 ± 2.47), then in social (14.45 ± 3.10) and physical (13.97 ± 2.62). The lowest score was noted in the environmental domain (13.63 ± 2.54). On the anxiety scale, 187 (37.63%) respondents showed a borderline state, 186 (37.42%) reported no disorders in this respect but 124 (24.95%) had clear signs of anxiety. In terms of depression, 304 (61.17%) did not report any disorders, 111 (22.33%) showed borderline symptoms and 82 (16.50%) noted clear signs of depression (Table 2).

### 3.3. Occupational Burnout, Quality of Life and the Level of Anxiety and Depression vs. Profession

Emotional exhaustion was found significantly higher in the nursing staff and ‘other’ medical workers than in the physiotherapists (*p* = 0.015). Similarly, depersonalization was much stronger in the nursing staff and ‘other’ medical workers than in the physiotherapists or midwives (*p* = 0.023). On the contrary, job satisfaction was higher in the physiotherapists than among any other professions examined for the study (*p* < 0.001). The nursing staff reported definitely worse results at QOL perception (*p* = 0.003) and the psychological domain of QOL (*p* = 0.001) than any other professions. The self-perception of health condition was at a much better level in the physiotherapists than the nursing or midwifery staff (*p* < 0.001). What is more, it was higher in other professions than in the nursing staff (*p* < 0.001). The QOL in the physical domain was much higher in the ‘other’ medical workers than in other groups (*p* < 0.001). The social domain of the QOL was much higher in the representatives of ‘other’ workers than in the nursing staff as well (*p* = 0.02). The levels of anxiety and depression were much higher in the nursing staff than in the midwives or physiotherapists (*p* < 0.001) and, interestingly, they were higher in the ‘other’ professions than in physiotherapists (*p* < 0.001) (Table 3).

### 3.4. Emotional Exhaustion–Regression Analysis

The indirect predictors of emotional exhaustion were anxiety (*p* < 0.001), depression (*p* < 0.001), QOL perception (*p* < 0.001), perception of health condition (*p* < 0.001) and the QOL domains: physical (*p* < 0.001), psychological (*p* < 0.001), environmental (*p* < 00.1) and social (*p* < 0.001). Moreover, emotional exhaustion was closely related to age (*p* = 0.034), material status (*p* = 0.004) and being provided PPE (*p* < 0.001). Depression intensity, the level of QOL in the physical domain, provided PPE by an employer and material status proved to be direct predictors of emotional exhaustion. In the multifactorial analysis, each point at the depression subscale (HADS) increased the level of emotional exhaustion by 0.546 points on average (regression parameter 0.546, 95% CI 0.224, 0.867). Each point at the subscale of the physical domain of QOL decreased the level of emotional exhaustion by 1.961 points on average (regression parameter −1.961, 95% CI −2.519, −1.404) and each point on the provided PPE scale decreased the exhaustion by 1.36 points on average (regression parameter −1.36, 95% CI −2.129, −0.591). Material status appeared to be a direct predictor as well, but during the multifactorial analysis it changed its character. According to the monofactorial analysis, an average, bad or very bad material status increased the level of emotional exhaustion by 4.941 points on average in comparison to a very good status. However, the multifactorial analysis showed that a good material status decreased the level of exhaustion by 3.669 points on average compared to a very good status (regression parameter −3.669, 95% CI −6.037, −1.301) (Table 4).

The R^2^ coefficient for the emotional exhaustion model was 0.534, which means that this model explains 53.4% of the variability in the emotional exhaustion scale. However, the remaining 46.6% depends on variables not included in the model and on random factors. 

### 3.5. Depersonalization–Regression Analysis

The monofactorial analysis revealed indirect predictors of depersonalization such as the level of anxiety (*p* < 0.001), depression (*p* < 0.001), QOL perception (*p* < 0.001), perception of health condition (*p* < 0.001), and the QOL in the physical (*p* < 0.001), psychological (*p* < 0.001), environmental (*p* < 0.001) and social (*p* < 0.001) domains. Depersonalization was also linked to age (*p* = 0.001), working in POZ (primary healthcare institutions) (*p* = 0.016), being provided PPE (*p* = 0.002), a nursing profession (*p* = 0.023) and ‘other’ medical professions (*p* = 0.025). The QOL in the physical and social domains, age, working in a primary healthcare clinic and the nursing profession appeared to be direct predictors of depersonalization. The multifactorial analysis showed that each point on the QOL physical domain subscale decreased the level of depersonalization by 0.471 points on average (regression parameter −0.471, 95% CI −0.8, −0.142), each point at the QOL social domain subscale decreased the level of depersonalization by 0.279 points on average (regression parameter −0.279, 95% CI −0.494, −0.064), each next year at the age scale decreased depersonalization by 0.081 points on average (regression parameter −0.081, 95% CI −0.131, −0.031), working at PHC increased depersonalization by 1.95 points on average (regression parameter 1.95, 95% CI 0.339, 3.561), and a nursing profession compared to being a midwife increased the level of depersonalization by 1.712 points on average (regression parameter 1.712, 95% CI 0.29, 3.134) (Table 5).

The R^2^ coefficient for the depersonalization model was 0.286, which means that this model explains 28.6% of the variation in the depersonalization scale. The remaining 71.4% depends on variables not included in the model and on random factors.

### 3.6. Job Satisfaction–Regression Analysis

The monofactorial analysis showed that indirect predictors of job satisfaction included the level of anxiety (*p* < 0.001), depression (*p* < 0.001), QOL perception (*p* < 0.001), perception of health condition (*p* < 0.001) as well as the QOL in physical (*p* < 0.001), psychological (*p* < 0.001) environmental (*p* < 0.001) and social (*p* < 0.001) domains. Job satisfaction was also related to age (*p* = 0.018), a good (*p* = 0.02), average, bad and very bad material status (*p* = 0.001), working at hospital different than COVID-19 or contagious (*p* = 0.011), being provided PPE (*p* < 0.001) getting through quarantine (*p* = 0.017), getting through SARS-CoV-2 infection (*p* = 0.008) and being a physiotherapist (*p* = 0.014). Depression, QOL perception in the psychological and social domains, being provided PPE by an employer and doing ‘other’ medical professions turned out to be direct predictors of job satisfaction. In the multifactorial analysis, each point on the depression subscale decreased the level of job satisfaction by 0.373 points on average (regression parameter −3.373, 95% CI −0.642, −0.104), each point at the QOL perception scale decreased job satisfaction by 1.132 points on average (regression parameter −1.132, 95% CI −2.162, −0.102), each point at the QOL psychological domain subscale increased the level of job satisfaction by 0.699 points on average (regression parameter 0.699, 95% CI 0.199, 1.199), each point at the QOL social domain subscale increased it by 0.358 points on average (regression parameter 0.358, 95% CI 0.052, 0.663) and each point at the QOL environmental domain subscale increased it by 0.472 points on average (regression parameter 0.472, 95% CI 0.045, 0.899). What is more, each point on the being provided PPE subscale increased the level of job satisfaction by 1.004 points on average (regression parameter 1.004, 95% CI 0.36, 1.648), while performing ‘other’ medical professions decreased job satisfaction by 3.084 points on average in comparison to being a midwife (regression parameter −3.084, 95% CI −5.783, −0.384) (Table 6).

The R^2^ coefficient for the job satisfaction model was 0.358, which means that this model explains 35.8% of the variation in the job satisfaction scale. The remaining 64.2% depends on the variables not included in the model and on random factors.

## 4. Discussion

The study presented above analysed the phenomenon of occupational burnout, the level of anxiety and depression, as well as quality of life among healthcare workers in the time of the COVID-19 pandemic. The research also aimed at establishing the factors significantly determining the level of occupational burnout in medical workers. It was shown that over three quarters (356; 71.63%) of the workforce displayed high or moderate levels of emotional exhaustion as well as low and moderate levels of job satisfaction (355; 71.43%). Additionally, two fifths (203; 40.85%) of the respondents experienced high or moderate levels of depersonalization. The staff examined for the study evaluated their QOL the highest in the psychological domain and the lowest in the environmental one. The group of 62.57% (*n* = 311) suffered anxiety disorders or its borderline states while 38.83% (*n* = 193) experienced clear depression disorders or their borderline states. The direct predictors of occupational burnout included escalating depression, the QOL domains, being provided PPE by an employer as well as such sociodemographic data as age, profession and material status.

### 4.1. Occupational Burnout, Quality of Life and the Level of Anxiety and Depression

The data collected in the study confirm the negative impact of stressors connected with the COVID-19 pandemic on healthcare workers in Poland. A high level of emotional exhaustion and low level of job satisfaction was found in over 40% of the respondents, with a high level of depersonalization in over 17%. In the systematic review of Gualano et al. (*n* = 12.596 HCW’s in intensive care units and emergency departments), high levels of emotional exhaustion ranged from 3.1% to 24.7%, high levels of depersonalization from 12.5% to 21.1%, and high levels of lacking personal accomplishments from 1.1% to 25% [39]. In comparison to Gualamo, the self-reported study noted higher percentages of emotionally exhausted and less satisfied medical workers. Similar findings were also reported by other researchers who examined European and worldwide medical staff. Spanish research conducted among healthcare workers during the COVID-19 pandemic reported a group of 41% workers who suffered from high emotional exhaustion, a group of 15.2% who felt high depersonalization and 8.4% who showed low levels of personal accomplishment. Moreover, a total of 56.6% of health workers presented symptoms of post-traumatic stress disorder, 58.6% anxiety disorder, and 46% depressive disorder [24]. A huge psychophysical influence of the pandemic on the medical workforce was also presented in the research conducted in Italy by Barello et al. Italian medical workers reported high psychological pressure linked to work, emotional exhaustion and somatic symptoms [23]. In the study of Khasne et al., conducted among 2026 healthcare workers in India, greater than half of the respondents (1.069; 52.8%) had pandemic related burnout [40]. The references available in this respect contain a few studies that prove the impact of the COVID-19 pandemic on the level of occupational burnout among physiotherapists [41,42]. The study carried out during the pandemic among Portuguese physiotherapists showed that more than 40% of them experienced personal and work related burnout and 25% patient related burnout, with resilience, depression and stress having a relevant role in the three burnout dimensions [41]. The examination conducted by Pniak et al. (2021), in south-east Poland with the use of MBI questionnaire among professionally active physiotherapists working in hospitals, showed that physiotherapists presented high burnout rates in all three dimensions: emotional exhaustion (Mean 32.31; CI 29.47–35.15), depersonalization (Mean 16.25; CI 14.48–18.03) and personal accomplishment (Mean 26.25; CI 24.41–28.10) [42]. The physiotherapists examined for the self-reported study achieved the following average results, respectively: 21.78 ± 12.64 (moderate level) of emotional exhaustion, 5.87 ± 5.32 (low level) of depersonalization and 36.39 ± 8.26 (moderate level) of personal accomplishment. The differences in the results collected by Pniak et al. and the authors might stem from the fact that the physiotherapists examined in the study were employed by healthcare institutions other than hospitals (only 28.31% of them worked in hospitals).

What is more, the self-reported study proved that the nursing staff and ‘other’ medical workers (physicians and paramedics) showed significantly higher levels of emotional exhaustion and depersonalization than other groups examined. In addition, taking into account all the respondents, the anxiety disorders were found in 24.95%, and clear depression disorders in 16.50% of the sample group. In the national study of Young et al. (*n* = 1.685), nearly half of the HCWs reported serious psychiatric symptoms, such as depression, anxiety and suicidal ideation, during the COVID-19 pandemic [43]. Pappa et al. conducted a systematic review and analysed existing evidence on the prevalence of depression and anxiety among healthcare workers during the COVID-19 outbreak. They showed that pooled prevalence of anxiety was 23.2% and the prevalence rate of depression was 22.8% [44].

The results are convergent with the self-reported study to a high extent. With respect to anxiety and depression symptoms, the self-reported physiotherapists were in a much more favourable situation. The levels of anxiety and depression were much higher in the nursing staff compared to the midwives and physiotherapists and they were also much higher in the representatives of ‘other’ medical professions than in the physiotherapists. The results clearly show that the nursing staff and ‘other’ medical workers are in a much more disadvantageous situation, of all the groups, in terms of escalating anxiety and depression. The differences in occupational burnout intensity or anxiety and depression levels among medical workers representing various professions might result from the character or length and frequency of contacts with COVID-19 patients. A similar opinion was confirmed by Luceño-Moreno et al. [24]. High stress and occupational exhaustion levels as well as moderate depression were reported among nurses in Turkey [25]. A large scale research that examined injury indicators and occupational burnout among nurses during the COVID-19 pandemic was also conducted in China. In the survey, 13.3% reported trauma, there were moderate degrees of emotional exhaustion, and 39.3% experienced post-traumatic growth [26]. The self-reported results, together with the others presented above, require precise consideration, as emotional stress affects health condition longitudinally, including PTSD occurrence. The correlation between the factors was proven by Johnson et al., who conducted research among HCPs working directly with COVID-19 patients [45].

The regression analysis conducted in our study showed that depression was a direct predictor of emotional exhaustion and job satisfaction and an indirect predictor of depersonalisation. The severity of depression adversely affected all three components of job burnout. There is the widespread but controversial opinion that burnout is a prodromal syndrome of depression [46]. The study conducted among nurses by Pachi et al. also highlighted that burnout is a factor influencing depression. In this study, regression evidenced that 43.7% of the variation in the BDI (Beck’s Depression Inventory) rating was explained by the CBI (Copenhagen Burnout Inventory) [17]. Kaschka et al. also indicated that burnout is likely to be a risk factor for developing depression [47]. As we indicated in the introduction, there is a lot of controversy among researchers over the overlap of depression and burnout. It is certainly advisable to conduct further research in this area.

However, it is difficult to determine the impact of the COVID-19 pandemic on the level of occupational burnout of health care workers and on the level of anxiety and depression without previous studies before the pandemic. Psychologists emphasize that people employed in social organizations are exposed to chronic stress and burnout syndrome caused by the need to maintain close contact with another person (patient, client), but with a simultaneous feeling of dissatisfaction and lack of connection. However, emotional exhaustion, depersonalization, and a sense of diminished personal achievement can also be attributed to other external factors. One of them is poor living and working conditions. According to the Euro Health Consumer Index, the Polish health care system is one of the worst among the 35 surveyed countries. Poland was ranked thirty-first out of 35 assessed countries, scoring only 564 out of 1000 points. Poland performed particularly poorly in terms of access to health care and access to medications, waiting time for treatment, the scope of services provided, as well as patient rights and access to information [48].

Bureaucracy, which causes medical staff to spend a lot of time keeping medical records instead of focusing on actual patient care, and the low salaries of interns, residents, nurses and physiotherapists further increase the ineffectiveness of healthcare in Poland. In the research on Głębocka carried out in Poland, only doctors assessed the health care system as relatively good [49]. Polish researchers show a relatively high level of occupational burnout among health care workers, even before the pandemic. Makara-Studzińska et al. study the degree of occupational burnout among doctors using the Link Burnout Questionnaire (LBQ). The results of measurements of the four aspects of occupational burnout indicate an average or high level of burnout among Polish doctors. Every second doctor participating in the study declared a high degree of occupational burnout in each of the aspects [50]. The research of Polish nurses showed that about 27% of nurses reported occupational burnout and over 28% burnout in contacts with patients [51]. A review study by Mikołajewska et al. showed that physical therapists also experienced burnout often and very often (13.8–65.1%), ranging from mild to high [52]. The COVID-19 pandemic has certainly had an impact on the level of burnout among health care workers. However, due to the relatively high level of burnout in healthcare professionals that also existed before the pandemic, one should be careful in drawing firm conclusions.

Depression and close contact with COVID-19 patients negatively affect medical staff’s quality of life [53]. Therefore, analysing QOL in this group was one of the aims of the research. The average QOL assessment among the examinees amounted for 3.67 ± 0.79 points, which means the respondents evaluated their QOL as good or average (not too good and not too bad). Moreover, the average value for the perception of their health condition was 3.54 ± 0.83, which means they evaluated their health as between good and average. The highest scores were achieved in the psychological domain of QOL, while the lowest was in the environmental one. It ought to be highlighted, additionally, that the nursing staff had much lower results than other professions examined in the study and significantly lower in the social domain of QOL. Similar findings were noticed in perceptions of health condition, which was lower in the nurses than in the physiotherapists. As it may be observed, the nursing staff was found to be in a much more disadvantageous situation compared to other professions. In the study of Woon et al., conducted among Malaysian HCW’s, all domains of QoL were within the norms of the general population except for the social relationship QoL, which was lower than the norm [54].

The self-reported study’s score was 14.46 ± 2.47 in the psychological, 14.45 ± 3 in the physical, 13.97 ± 2.62 in the social, and 13.63 ± 2.54 in the environmental domains. In comparison to Maqsood et al., the self-reported results are much better in both aspects: the average QOL and perception of health condition. In the study of Maqsood et al., the mean overall quality of life score was 3.37 ± 0.97, general health was 3.66 ± 0.88, domains, i.e., the physical was 11.67 ± 2.16, the psychological was 13.08 ± 2.14, the social was 13.22 ± 3.31 and the environment was 12.38 ± 2.59 [55].

Such a difference may stem from the fact that the researchers mentioned above conducted their research at hospital wards, such as healthcare staff of intensive care units and emergency units, where the workers have come into frequent contact with COVID-19 patients, serious conditions and deaths, while our respondents constituted a more divergent group in terms of a workplace (hospital workers and open healthcare workers), which might have its impact on a better quality of their lives.

### 4.2. Factors Determining the Level of Occupational Burnout

The results collected for the study clearly prove that depression stems from two areas of occupational burnout. Each point on the depression subscale significantly decreased the level of job satisfaction and increased the level of emotional exhaustion. This means that the higher the intensity of depression symptoms, the lower the job satisfaction and the higher the emotional exhaustion. Therefore, anxiety and depression, linked to the pandemic, might only enhance occupational burnout. Similar findings were reported among Australian and Spanish healthcare workers, in whom occupational burnout was clearly related to the levels of anxiety and depression [24,56].

The self-reported study also presented that the physical domain of QOL has a great impact on occupational burnout as well. Each point on the physical domain of QOL scale decreased both emotional exhaustion and depersonalization. It is essential to mention that the physical domain of the WHOQOOL questionnaire includes such aspects as daily routine activities, drug and treatment dependence, vigour, tiredness, mobility, pain and discomfort, recreation, sleep and capacity for work [57]. Therefore, a healthcare worker needs to take all these aspects into account daily while trying to compensate for emotional exhaustion and depersonalization. The social domain also appeared to be equally important for the levels of depersonalization and job satisfaction. As each point on its subscale decreased the level of depersonalization and increased the level of job satisfaction, it is essential to care for personal relationships, social support and sexual activity to prevent occupational burnout. All the elements mentioned above, after all, constitute the social domain of WHOQOOL questionnaire [57].

The organizational reasons affecting the level of occupational burnout include material status, being provided PPE by an employer and the workplace itself. Material status appeared to be a direct predictor of emotional exhaustion. What is more, a low material status indirectly influenced job satisfaction as well. The issue of earnings in the healthcare sector has been present in Poland for many years. According to GUS (Central Statistical Office), the average wage in the healthcare and social support sector was PLN 5 371.73 (about EUR 1191). In addition, it was only a bit higher than the average wage generally (PLN 5 167.47, about EUR 1146) [58]. Such unsatisfactory earnings of highly specialized staff lead to seeking additional employment which, in consequence, results in work overload (in Poland, over 60% of doctors and approximately 30–40% of nurses work at more than one medical institution). This low work prestige and an unsatisfactory level of earnings is related mostly to Polish nurses [59]. If the situation is worsened by the pandemic overload, it may easily lead to the decline in job satisfaction.

The self-reported study found being provided PPE by an employer as a factor that decreases emotional exhaustion and depersonalization as well as increasing job satisfaction. Using PPE and proper equipment is indispensable in order to take adequate care of COVID-19 patients and reduce the possibility of staff infection. Similar results were also reported by other researchers. Spanish healthcare workers provided with PPE declared fewer anxiety and depression symptoms as well as lower occupational burnout [20]. According to Shanafelt et al., access to PPE was a major source of anxiety among their healthcare workers [19]. The identification of anxiety sources may allow managers and healthcare organizers to develop an attitude on how to deal with such issues and ensure proper support for all healthcare workers.

Individual factors affecting occupational burnout include age and gender as well. Age was an indirect predictor of emotional exhaustion, depersonalization and job satisfaction. The older the worker, the lower the emotional exhaustion and depersonalization and the higher the level of job satisfaction. It might be concluded that older age together with years spent working in a profession result in better coping with daily professional challenges. Murat et al., who conducted their research among Turkish nurses, achieved similar results to the self-reported ones. Younger staff, characterised by less job experience, presented higher levels of occupational burnout [25]. The self-reported study did not reveal any significance of gender in this respect. However, there were contrary results reported by other authors. Women experienced higher emotional exhaustion and depersonalization during the pandemic [24]. Portuguese female physiotherapists revealed higher scores concerning occupational burnout [41]. What is more, Luceno-Moreno et al. indicated that being female was significantly and positively related to anxiety and depression [24]. De Kock et al. suggest that female nurses with close contact with COVID-19 patients may have the most to gain from efforts aimed at supporting psychological well-being [4].

The COVID-19 infection and the need for quarantine might also considerably influence the level of occupational burnout. The research proved that both of them indirectly decreased the level of job satisfaction. COVID-19 infection, isolation, probable hospitalization and home transmission, as well as the fear of infecting relatives, constitute incredibly stressful factors. Therefore, job satisfaction might have been lowered among those healthcare workers who underwent COVID-19 infection and related quarantine. Higher occupational burnout was found among Turkish nurses who were COVID-19 positive [25]. Similarly, American medics who suffered from COVID-19 reported higher occupational burnout, depression and anxiety compared to healthy workers [60]. The results highlight an urgent need for physical and psychological support among healthcare workers. The self-reported study examined healthcare workers in a few groups such as nurses, midwives, physiotherapists and ‘other’ medical professions. The analysis revealed that working as a nurse increased the level of depersonalization in comparison to working as a midwife. The respondents representing ‘other’ medical professions reported lower job satisfaction levels than midwives. Therefore, it might be concluded that working as a midwife was most favourable during the pandemic, as this staff was subjected to occupational burnout the least. Other researchers reported diversified findings. Buselli et al. proved equal occupational burnout among physicians, nurses and healthcare assistants [61]. However, the systematic review of Gualano et al. reveals that nurses are at the highest risk of occupational burnout of all healthcare workers [39], which partly corresponds with our results. The research by Dobson et al. reported indicators of occupational burnout, such as depression and anxiety, as different for different professions–senior medical practitioners reported the lowest levels of psychological distress compared to junior medical practitioners, nurses or allied health practitioners [56]. Each group of healthcare professionals has different competences and duties, thus, some fluctuations in occupational burnout levels might occur.

### 4.3. Limitations of the Study

The strongest point of the study lies in the fact that it collected data from medical workers from most voivodships in Poland, including cities and villages. The results may indicate the issue of occupational burnout as present among healthcare workers all over the country regardless of any specific region or place of residence. They also imply that medical staff at each level and profession are susceptible to high levels of occupational burnout during the COVID-19 pandemic.

The research, however, has its limitations as well. The levels of occupational burnout, anxiety and depression had not been examined among the same workers before the pandemic. Without this control group, it is impossible to conclude if the level of burnout was not caused by other prepandemic factors present among the medical staff. Another limitation might be related to its cross-sectional character. The pandemic has not ended and its impact on the mental health of medical workers still continues. The longitudinal study would be legitimate to evaluate the aspects examined in the study in the proper time perspective. The next limitation may result from the usage of the internet questionnaire distributed by email and social media, which are inaccessible to some medical workers. However, the method allowed for data collection in a short period of time, which was convenient during the pandemic. Moreover, we have collected data from 497 subjects. We are not sure why the rest of the survey participants did not complete the survey, even though they initially expressed interest in the topic of our research. We can only assume that the difficult time of the pandemic and the excessive workload of healthcare workers could discourage the respondents from filling in a fairly extensive package of questions (the questionnaires contained a total of 80 questions). However, in order to thoroughly research the phenomenon of burnout from the perspective we are interested in, we decided that the tools we chose were necessary in the study. Further research, with a larger number of respondents, is needed to fully assess the impact of the COVID 19 effect on occupational burnout. Furthermore, more female workers definitely took part in the research as compared to male ones. Other research confirms this limitation, too [24,62]. In this case, the reason comes from the fact that most nursing and midwifery posts in Poland are held by women. Finally, physicians and paramedics were not assigned to two different groups, due to a small number of volunteers and were appointed to one group named ‘other’ medical professions. Some further research ought to take all the limitations mentioned above into serious consideration.

## 5. Conclusions

High and moderate levels of emotional exhaustion as well as low and moderate levels of job satisfaction were observed among medical staff. Part of them revealed high and moderate levels of depersonalization as well. The COVID-19 pandemic constitutes a new challenge for all healthcare workers, and decision-makers should pay more attention to the possibility of the initiation or worsening of burnout syndrome and the symptoms of anxiety and depression among healthcare professionals. The occupational burnout prophylaxis during the pandemic ought to be focused on early recognition and treatment of depression like symptoms, ensuring PPE and satisfying earnings, because escalating depression, material status and ensuring PPE are direct predictors of occupational burnout. Each worker should also pay great attention to such every day wellbeing aspects as vigour, tiredness, mobility, pain and discomfort, recreation, sleep and capacity for work, as well as social support and sexual activity, in order to compensate for emotional exhaustion and depersonalization as well as to increase job satisfaction. This all stems from the fact that the physical and social domains of QOL, together with the aspects mentioned above, could be determinants of occupational burnout.

Nursing staff appeared to be in a quite unfavourable situation compared to all the professions discussed above, in terms of levels of emotional exhaustion and depersonalization, perception of the QOL in the psychological domain, and levels of anxiety and depression. Therefore, the prophylactic guidance in terms of occupational burnout ought to be particularly applied to nursing staff. Nevertheless, it is necessary to develop preventive guidelines and implement psychological and administrative interventions aimed at reducing the phenomenon of professional burnout, depression and raising the quality of life in relation to all the representatives of the professions we studied. Taking into account the fact that one of the problems of the Polish health care system is the shortage of medical staff, in particular doctors and nurses, it is necessary for the decision-makers to take actions aimed at improving the existing situation.

## Figures and Tables

**Table 1 ijerph-19-03634-t001:** The characteristics of the respondents.

Parameter		Occupation				
		Midwife—A (*n* = 82)	Nurse/Male Nurse—B (*n* = 240)	Physiotherapist—C (*n* = 106)	Other Medical Professions—D (*n* = 69)	Total (*n* = 497)
Age (years)	^1^ M ± ^2^ SD	37.43 ± 10.43	42.4 ± 10.54	37.19 ± 9.61	39.48 ± 10.94	40.06 ± 10.62
	median	36	45	37	38	41
	^3^ Q1–^4^ Q3	29–45	33.75–50	29–45	31–47	30–49
Gender	women	81 (98.78%)	233 (97.08%)	84 (79.25%)	44 (63.77%)	442 (88.93%)
	men	1 (1.22%)	7 (2.92%)	22 (20.75%)	25 (36.23%)	55 (11.07%)
The number of people living in the household	M ± SD	3.05 ± 1.39	2.78 ± 1.2	3.13 ± 1.47	2.78 ± 1.24	2.9 ± 1.3
	median	3	3	3	3	3
	Q1–Q3	2–4	2–4	2–4	2–4	2–4
Education	secondary	4 (4.88%)	50 (20.83%)	4 (3.77%)	8 (11.59%)	66 (13.28%)
	bachelor degree	28 (34.15%)	74 (30.83%)	18 (16.98%)	10 (14.49%)	130 (26.16%)
	master degree	48 (58.54%)	112 (46.67%)	71 (66.98%)	48 (69.57%)	279 (56.14%)
	PhD	2 (2.44%)	4 (1.67%)	13 (12.26%)	3 (4.35%)	22 (4.43%)
Place of residence	city	61 (74.39%)	174 (72.50%)	74 (69.81%)	63 (91.30%)	372 (74.85%)
	village	21 (25.61%)	66 (27.50%)	32 (30.19%)	6 (8.70%)	125 (25.15%)
The financial status of the family	very good	12 (14.63%)	32 (13.33%)	15 (14.15%)	20 (28.99%)	79 (15.90%)
	good	51 (62.20%)	121 (50.42%)	43 (40.57%)	33 (47.83%)	248 (49.90%)
	average	19 (23.17%)	84 (35.00%)	41 (38.68%)	15 (21.74%)	159 (31.99%)
	bad	0 (0.00%)	2 (0.83%)	7 (6.60%)	1 (1.45%)	10 (2.01%)
	very bad	0 (0.00%)	1 (0.42%)	0 (0.00%)	0 (0.00%)	1 (0.20%)
Life in a stable relationship	No	15 (18.29%)	51 (21.25%)	28 (26.42%)	14 (20.29%)	108 (21.73%)
	Yes	67 (81.71%)	189 (78.75%)	78 (73.58%)	55 (79.71%)	389 (78.27%)
Number of workplaces	M ± SD	1.44 ± 0.57	1.41 ± 0.61	1.47 ± 0.72	1.72 ± 0.75	1.47 ± 0.65
	median	1	1	1	2	1
	Q1–Q3	1–2	1–2	1–2	1–2	1–2
Place of work	Primary healthcare clinic	8 (9.76%)	38 (15.83%)	15 (14.15%)	13 (18.84%)	74 (14.89%)
	Specialist clinic	7 (8.54%)	22 (9.17%)	22 (20.75%)	15 (21.74%)	66 (13.28%)
	Speciality Hospital for Infectious Diseases	4 (4.88%)	23 (9.58%)	2 (1.89%)	6 (8.70%)	35 (7.04%)
	Other hospitals	69 (84.15%)	156 (65.00%)	28 (26.42%)	36 (52.17%)	289 (58.15%)
	Long term or palliative care home facilities	1 (1.22%)	10 (4.17%)	8 (7.55%)	4 (5.80%)	23 (4.63%)
	Inpatient long-term or palliative care facilities	0 (0.00%)	16 (6.67%)	4 (3.77%)	7 (10.14%)	27 (5.43%)
	Other	19 (23.17%)	39 (16.25%)	57 (53.77%)	29 (42.03%)	144 (28.97%)
Provision of personal protective equipment at the workplace	M ± SD	3.24 ± 1.05	3.66 ± 1.09	3.66 ± 1.24	3.8 ± 1.11	3.61 ± 1.13
	median	3	4	4	4	4
	Q1–Q3	3–4	3–4	3–5	3–5	3–4
Quarantine	No	69 (84.15%)	164 (68.33%)	95 (89.62%)	42 (60.87%)	370 (74.45%)
	Yes	13 (15.85%)	76 (31.67%)	11 (10.38%)	27 (39.13%)	127 (25.55%)
Infection with SARS- COV 2	No	73 (89.02%)	209 (87.08%)	102 (96.23%)	58 (84.06%)	442 (88.93%)
	Yes	9 (10.98%)	31 (12.92%)	4 (3.77%)	11 (15.94%)	55 (11.07%)

Legend: ^1^ mean, ^2^ standard deviation, ^3^ Q1—first quartille,^4^ Q3—third quartille.

**Table 2 ijerph-19-03634-t002:** Occupational burnout quality of life and level of anxiety and depression in healthcare workers during the pandemic.

WHOQoL BREF	*n*	M ^1^	SD ^2^	Median	Min	Max	Q1 ^3^	Q3 ^4^
Physical Domain	497	13.97	2.62	14	7	20	12	16
Psychological domain	497	14.46	2.47	15	4	20	13	16
Social domain	497	14.45	3.10	15	4	20	12	16
Environmental domain	497	13.63	2.54	14	5	20	12	16
Intensity			Hospital Anxiety Depression Scale					
			Anxiety				Depression	
No disorders			186 (37.42%)				304 (61.17%)	
Borderline state			187 (37.63%)				111 (22.33%)	
Disorders present			124 (24.95%)				82 (16.50%)	
Maslach Burnout Inventory								
Emotional exhaustion						*n*	%	
Points			Interpretation					
0–16			Low			141	28.37%	
17–26			Moderate			139	27.97%	
>26			High			217	43.66%	
Depersonalization						*n*	%	
Points			Interpretation					
0–6			Low			294	59.15%	
7–12			Moderate			116	23.34%	
>12			High			87	17.51%	
Lack of accomplishment						*n*	%	
points			Interpretation					
0–31			Low			205	41.25%	
32–38			Moderate			150	30.18%	
>38			High			142	28.57%	

Legend: ^1^ mean, ^2^ standard deviation, ^3^ Q1—first quartille, ^4^ Q3—third quartille.

**Table 3 ijerph-19-03634-t003:** Quality of life occupational burnout and the level of anxiety and depression vs. profession.

		Occupation				*p* ^5^
		Midwife—A (*n* = 82)	Nurse/Male Nurse—B (*n* = 240)	Physiotherapist—C (*n* = 106)	Other Medical Professions—D (*n* = 69)	
Maslach Burnout Inventory						
Emotional exhaustion	M ^1^ ± SD ^2^	24.74 ± 12.19	26.48 ± 12.38	21.78 ± 12.64	25.75 ± 13.34	*p* = 0.015 *
	median	23	25	21	26	
	Q1 ^3^–Q3 ^4^	16–32	16–37	11.25–28	15–34	B, D > C
Depersonalization	M ± SD	5.67 ± 5.39	7.41 ± 6.37	5.87 ± 5.32	7.87 ± 6.19	*p* = 0.023 *
	median	4	6	5	7	
	Q1–Q3	1–9	2–11	1–9.75	3–12	D, B > C, A
Lack of accomplishment	M ± SD	33.17 ± 7.43	31.68 ± 9.42	36.39 ± 8.26	31.12 ± 9.09	*p* < 0.001 *
	median	33.5	32	37	32	
	Q1–Q3	28–38	26–39	33–42	26–37	C > A, B, D
WHOQoLBREF						
QoL perception	M ± SD	3.78 ± 0.72	3.56 ± 0.77	3.71 ± 0.85	3.86 ± 0.83	*p* = 0.003 *
	median	4	4	4	4	
	Q1–Q3	3–4	3–4	3–4	3–4	D, A, C > B
Perception own health	M ± SD	3.54 ± 0.74	3.41 ± 0.82	3.77 ± 0.87	3.62 ± 0.88	*p* < 0.001 *
	median	4	4	4	4	
	Q1–Q3	3–4	3–4	3–4	3–4	C > A, B D > B
Physical domain	M ± SD	13.99 ± 2.53	13.53 ± 2.53	15 ± 2.59	13.88 ± 2.72	*p* < 0.001 *
	median	14	13	15	14	
	Q1-Q3	13–15	12–15	13–17	12–16	C > A, D, B
Psychological domain	M ± SD	14.7 ± 2.44	14.01 ± 2.43	15.07 ± 2.55	14.8 ± 2.26	*p* = 0.001 *
	median	15	14	15	15	
	Q1–Q3	13–16.75	13–16	14–17	13–17	C, D, A > B
Social domain	M ± SD	14.62 ± 3.51	14.09 ± 3.05	15.2 ± 2.9	14.35 ± 2.85	*p* = 0.02 *
	median	16	15	16	15	
	Q1–Q3	12–16	12–16	13–17	12–16	C > B
Environmental domain	M ± SD	13.68 ± 2.44	13.33 ± 2.58	14.08 ± 2.41	13.91 ± 2.62	*p* = 0.069
	median	14	14	14	14	
	Q1–Q3	12–16	12–15	12–16	12–16	
Hospital Anxiety Depression Scale						
Anxiety	M ± SD	8.41 ± 3.04	9.35 ± 3.21	7.5 ± 3.13	8.67 ± 3.28	*p* < 0.001 *
	median	8	9	7	8	
	Q1–Q3	6.25–10	7–11	5–9	7–10	B > A, C D > C
Depression	M ± SD	5.54 ± 4.04	6.92 ± 4.11	4.49 ± 4.29	6.33 ± 4.58	*p* < 0.001 *
	median	5	7	4	6	
	Q1–Q3	2–8	3–10	0.25–7.75	3–10	B > A, C D > C

Legend: ^1^ mean, ^2^ standard deviation, ^3^ Q1—first quartile, ^4^ Q3—third quartile, ^5^ Kruskal–Wallis test and posthoc analysis (Dunn test), * statistically significant dependence (*p* < 0.05).

**Table 4 ijerph-19-03634-t004:** Emotional exhaustion–mono- and multifactorial regression analysis.

Feature		Univariate Models				Multivariate Model			
		Parameter	95% CI		*p*	Parameter	95% CI		*p*
HADS: Anxiety		2.122	1.835	2.41	<0.001 *	0.237	−0.162	0.636	0.245
HADS: Depression		1.778	1.572	1.984	<0.001 *	0.546	0.224	0.867	0.001 *
WHOQoL-BREF: QoL perception		−5.035	−6.366	−3.704	<0.001 *	0.338	−0.891	1.567	0.59
WHOQoL-BREF: Perception own health		−5.6	−6.839	−4.361	<0.001 *	1.031	−0.289	2.351	0.126
WHOQoL-BREF: Physical domain		−3.131	−3.453	−2.809	<0.001 *	−1.961	−2.519	−1.404	<0.001 *
WHOQoL-BREF: Psychological domain		−2.918	−3.289	−2.547	<0.001 *	−0.286	−0.883	0.31	0.347
WHOQoL-BREF: Social domain		−1.954	−2.27	−1.639	<0.001 *	−0.279	−0.643	0.085	0.134
WHOQoL-BREF: Environmental domain		−2.645	−3.016	−2.274	<0.001 *	−0.283	−0.792	0.227	0.277
Age	[years]	−0.113	−0.217	−0.009	0.034 *	−0.082	−0.167	0.003	0.059
Gender	Women	ref.				ref.			
	Men	−2.637	−6.175	0.901	0.145	1.018	−1.842	3.877	0.486
Education	Secondary	ref.				ref.			
	Bachelor degree	−2.038	−5.779	1.703	0.286	−2.213	−5.109	0.683	0.135
	Master degree/Phd	0.129	−3.235	3.493	0.94	−0.743	−3.428	1.942	0.588
Place of residence	City	ref.				ref.			
	Village	−0.206	−2.77	2.357	0.875	−0.224	−2.209	1.761	0.825
The number of people living in the household		−0.556	−1.409	0.297	0.202	−0.497	−1.164	0.17	0.145
Material status	Very good,	ref.				ref.			
	good	0.118	−3.035	3.271	0.942	−3.669	−6.037	−1.301	0.003 *
	Average, bad, very bad	4.941	1.617	8.264	0.004 *	−2.014	−4.726	0.697	0.146
Life in a stable relationship	No	ref.				ref.			
	Yes	0.754	−1.942	3.451	0.584	1.079	−1.028	3.185	0.316
Number of workplaces		1.429	−0.27	3.128	0.1	1.189	−0.523	2.902	0.174
Place of work: primary healthcare clinic (PHC)	No	ref.				ref.			
	Yes	2.974	−0.14	6.087	0.062	0.806	−1.924	3.537	0.563
Place of work: specialist clinic	No	ref.				ref.			
	Yes	0.874	−2.403	4.151	0.601	0.573	−2.191	3.337	0.685
Place of work: speciality hospital for infectious siseases	No	ref.				ref.			
	Yes	1.04	−3.306	5.386	0.639	0.063	−3.59	3.716	0.973
Place of work: other hospitals	No	ref.				ref.			
	Yes	1.338	−0.914	3.59	0.245	−1.22	−3.836	1.395	0.361
Place of work: other	No	ref.				ref.			
	Yes	−1.467	−3.915	0.981	0.241	−1.369	−4.015	1.278	0.311
Place of work: long term or palliative care home facilities Inpatient long term or palliative care facilities	No	ref.				ref.			
	Yes	1.314	−2.523	5.15	0.502	−0.268	−3.681	3.146	0.878
Provision of personal protective equipment at the workplace		−3.02	−3.971	−2.068	<0.001 *	−1.36	−2.129	−0.591	0.001 *
Quarantine	No	ref.				ref.			
	Yes	−0.471	−3.02	2.079	0.718	−1.533	−3.747	0.681	0.175
Infection with SARS-CoV-2	No	ref.				ref.			
	Yes	2.658	−0.88	6.196	0.141	−0.03	−3.108	3.049	0.985
Occupation	Midwife	ref.				ref.			
	Nurse/male nurse	1.735	−1.409	4.88	0.28	0.06	−2.35	2.47	0.961
	Physiotherapist	−2.961	−6.576	0.655	0.109	−0.622	−3.63	2.386	0.685
	Other medical occupation	1.01	−3.007	5.026	0.622	−0.464	−3.685	2.758	0.778

Legend: * statistically significant dependence (*p* < 0.05).

**Table 5 ijerph-19-03634-t005:** Depersonalization–mono- and multifactorial regression analysis.

Feature		Univariate Models				Multivariate Model			
		Parameter	95% CI		*p*	Parameter	95% CI		*p*
Anxiety		0.58	0.425	0.735	<0.001 *	−0.112	−0.347	0.124	0.353
Depression		0.533	0.419	0.647	<0.001 *	0.144	−0.046	0.333	0.138
QoL perception		−1.461	−2.117	−0.805	<0.001 *	0.34	−0.385	1.065	0.358
Perception own health		−1.842	−2.456	−1.227	<0.001 *	−0.096	−0.875	0.683	0.809
Physical domain		−0.951	−1.135	−0.767	<0.001 *	−0.471	−0.8	−0.142	0.005 *
Psychological domain		−0.973	−1.17	−0.776	<0.001 *	−0.25	−0.602	0.102	0.164
Social domain		−0.678	−0.839	−0.518	<0.001 *	−0.279	−0.494	−0.064	0.011 *
Environmental domain		−0.755	−0.953	−0.557	<0.001 *	−0.062	−0.363	0.238	0.685
Age	[years]	−0.088	−0.137	−0.039	0.001 *	−0.081	−0.131	−0.031	0.002 *
Gender	Women	ref.				ref.			
	Men	0.976	−0.711	2.664	0.257	1.679	−0.009	3.366	0.052
Education	Secondary	ref.				ref.			
	Bachelor degree	0.924	−0.861	2.709	0.311	0.495	−1.214	2.204	0.57
	Master degree/Phd	0.919	−0.686	2.525	0.262	0.431	−1.154	2.015	0.595
Place of residence	City	ref.				ref.			
	Village	−0.517	−1.738	0.704	0.407	−0.285	−1.456	0.887	0.634
The number of people living in the household		−0.322	−0.728	0.085	0.121	−0.297	−0.691	0.097	0.14
Material status	Very good,	ref.				ref.			
	good	−0.018	−1.542	1.507	0.982	−0.843	−2.24	0.555	0.238
	Average, bad, very bad	0.835	−0.772	2.442	0.309	−0.927	−2.527	0.673	0.257
Life in a stable relationship	No	ref.				ref.			
	Yes	0.613	−0.671	1.897	0.35	1.193	−0.05	2.437	0.06
Number of workplaces		0.59	−0.22	1.401	0.154	0.172	−0.839	1.182	0.739
Place of work: primary healthcare clinic (PHC)	No	ref.				ref.			
	Yes	1.833	0.352	3.313	0.016 *	1.95	0.339	3.561	0.018 *
Place of work: specialist clinic	No	ref.				ref.			
	Yes	−0.187	−1.749	1.375	0.815	−0.185	−1.816	1.446	0.824
Place of work: speciality hospital for infectious diseases	No	ref.				ref.			
	Yes	1.688	−0.378	3.755	0.11	1.855	−0.3	4.011	0.092
Place of work: other hospitals	No	ref.				ref.			
	Yes	0.849	−0.223	1.921	0.121	1.157	−0.386	2.701	0.142
Place of work: other	No	ref.				ref.			
	Yes	−1.131	−2.296	0.033	0.057	−0.258	−1.819	1.304	0.747
Place of work: long term or palliative care home facilities Inpatient long term or palliative care facilities	No	ref.				ref.			
	Yes	1.185	−0.641	3.011	0.204	1.414	−0.601	3.428	0.17
Provision of personal protective equipment at the workplace		−0.74	−1.206	−0.274	0.002 *	−0.342	−0.795	0.112	0.141
Quarantine	No	ref.				ref.			
	Yes	−0.149	−1.364	1.066	0.81	−0.668	−1.974	0.638	0.317
Infection with SARS-CoV-2	No	ref.				ref.			
	Yes	0.649	−1.04	2.338	0.452	−0.075	−1.891	1.742	0.936
Occupation	Midwife	ref.				ref.			
	Nurse/male nurse	1.742	0.243	3.241	0.023 *	1.712	0.29	3.134	0.019 *
	Physiotherapist	0.197	−1.526	1.921	0.823	1.57	−0.205	3.345	0.084
	Other medical occupation	2.199	0.284	4.113	0.025 *	1.713	−0.188	3.614	0.078

Legend: * statistically significant dependence (*p* < 0.05).

**Table 6 ijerph-19-03634-t006:** Job satisfaction–mono- and multifactorial regression analysis.

Feature		Univariate Models				Multivariate Model			
		Parameter	95% CI		*p*	Parameter	95% CI		*p*
Anxiety		−1.027	−1.254	−0.8	<0.001 *	0.303	−0.031	0.637	0.076
Depression		−0.973	−1.136	−0.809	<0.001 *	−0.373	−0.642	−0.104	0.007 *
QoL perception		2.249	1.267	3.23	<0.001 *	−1.132	−2.162	−0.102	0.032 *
Perception own health		3.091	2.18	4.003	<0.001 *	−0.559	−1.664	0.547	0.323
Physical domain		1.602	1.334	1.869	<0.001 *	0.455	−0.012	0.922	0.057
Psychological domain		1.716	1.432	2	<0.001 *	0.699	0.199	1.199	0.006 *
Social domain		1.137	0.901	1.373	<0.001 *	0.358	0.052	0.663	0.022 *
Environmental domain		1.575	1.294	1.855	<0.001 *	0.472	0.045	0.899	0.031 *
Age	[years]	0.09	0.016	0.164	0.018 *	0.05	−0.021	0.122	0.165
Gender	Women	ref.				ref.			
	Men	0.742	−1.787	3.271	0.565	−0.804	−3.2	1.592	0.511
Education	Secondary	ref.				ref.			
	Bachelor degree	−0.48	−3.148	2.187	0.724	−0.11	−2.537	2.316	0.929
	Master degree/Phd	1.159	−1.24	3.558	0.344	0.899	−1.351	3.148	0.434
Place of residence	City	ref.				ref.			
	Village	−0.803	−2.631	1.025	0.39	−1.363	−3.026	0.301	0.109
The number of people living in the household		0.197	−0.413	0.806	0.527	0.226	−0.333	0.785	0.428
Material status	Very good,	ref.				ref.			
	good	−2.697	−4.962	−0.432	0.02 *	−0.557	−2.541	1.427	0.583
	Average, bad, very bad	−3.908	−6.295	−1.521	0.001 *	−0.061	−2.333	2.211	0.958
Life in a stable relationship	No	ref.				ref.			
	Yes	−0.548	−2.471	1.376	0.577	−1.464	−3.229	0.301	0.105
Number of workplaces		0.663	−0.551	1.877	0.285	0.782	−0.653	2.217	0.286
Place of work: primary healthcare clinic (PHC)	No	ref.				ref.			
	Yes	0.733	−1.495	2.961	0.519	0.357	−1.931	2.644	0.76
Place of work: specialist clinic	No	ref.				ref.			
	Yes	1.572	−0.763	3.906	0.188	0.38	−1.936	2.696	0.748
Place of work: speciality hospital for infectious diseases	No	ref.				ref.			
	Yes	−2.788	−5.88	0.304	0.078	−2.262	−5.323	0.799	0.148
Place of work: other hospitals	No	ref.				ref.			
	Yes	−2.087	−3.685	−0.489	0.011 *	−1.353	−3.545	0.838	0.227
Place of work: other	No	ref.				ref.			
	Yes	1.464	−0.281	3.209	0.101	−0.352	−2.569	1.865	0.756
Place of work: long term or palliative care home facilities Inpatient long term or palliative care facilities	No	ref.				ref.			
	Yes	1.077	−1.66	3.813	0.441	0.689	−2.171	3.549	0.637
Provision of personal protective equipment at the workplace		1.849	1.163	2.535	<0.001 *	1.004	0.36	1.648	0.002 *
Quarantine	No	ref.				ref.			
	Yes	−2.209	−4.018	−0.4	0.017 *	−0.148	−2.003	1.707	0.876
Infection with SARS-CoV-2	No	ref.				ref.			
	Yes	−3.429	−5.94	−0.917	0.008 *	−1.809	−4.389	0.77	0.17
Occupation	Midwife	ref.				ref.			
	Nurse/male nurse	−1.496	−3.71	0.719	0.186	−1.289	−3.309	0.73	0.211
	Physiotherapist	3.216	0.67	5.762	0.014 *	0.393	−2.128	2.913	0.76
	Other medical occupation	−2.055	−4.883	0.773	0.155	−3.084	−5.783	−0.384	0.026 *

Legend: * statistically significant dependence (*p* < 0.05).

## Data Availability

The data presented in this study are available on request from the corresponding author.

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
