# Peer review of "Predictors of the Occupational Burnout of Healthcare Workers in Poland during the COVID-19 Pandemic: A Cross-Sectional Study"

_ijerph, 2022, doi:10.3390/ijerph19063634_

Round 1

Reviewer 1 Report

The first word in the title of the article should read "predictors"

The article presented above analyzed the phenomenon of occupational burnout, the level of anxiety and depression as well as quality of life among healthcare workers in the time of the COVID-19 pandemic. The research also aimed at establishing the factors significantly determining the level of occupational burnout in medical workers. Research is relevant and represents an important direction in public health.

I want to congratulate the authors on a current and complete publication.

Author Response

Dear Editor

We would like to sincerely thank the Editorial Board and the Reviewer of your esteemed International Journal of Environmental Research and Public for their positive feedback and constructive recommendations improving our paper entitled: Predictors of occupational burnout of healthcare workers in Poland during the COVID-19 pandemic: a cross- sectional study (Manuscript ID: ijerph-1626225).  In this first round of revision, we focused our efforts strongly on the points made in your letter. We would like to respond to this opinion based on our careful revision, point by point, as you can see in the table below. Accordingly, the final version of the manuscript text includes the all necessary modifications and improvements.

Author's Reply to the Review Report:

REVIEWER #1

COMMENTS

AUTHORS’ REPLY

1.      The first word in the title of the article should read "predictors"

The first word in the title has been corrected.

2.      The article presented above analyzed the phenomenon of occupational burnout, the level of anxiety and depression as well as quality of life among healthcare workers in the time of the COVID-19 pandemic. The research also aimed at establishing the factors significantly determining the level of occupational burnout in medical workers. Research is relevant and represents an important direction in public health. I want to congratulate the authors on a current and complete publication.

The authors thank the reviewer for a positive comment regarding the subject taken.

We very much hope that our revisions appear comprehensive and proves helpful in obtaining a positive final decision accepting our paper for publication in your prestigeous journal. Awaiting your decision, I remain with kind regards

Authors

Reviewer 2 Report

I read the article entitled "Pedictors of occupational burnout of healthcare workers in  Poland during the COVID-19 pandemic: a cross-sectional study".

 Below are my remarks to the authors:

 Introduction:

-I would ask the authors to check again the definition proposed by the WHO for burnout, the WHO does not consider burnout a psychological syndrome but classifies it in the chapter "Problems associated with employment or unemployment". The definition given by WHO : «Burnout is a syndrome conceptualized as resulting from chronic workplace stress that has not been successfully managed. It is characterised by three dimensions: 1) feelings of energy depletion or exhaustion; 2) increased mental distance from one’s job, or feelings of negativism or cynicism related to one's job; and 3) a sense of ineffectiveness and lack of accomplishment. Burn-out refers specifically to phenomena in the occupational context and should not be applied to describe experiences in other areas of life» [  https://icd.who.int/browse11/l-m/en#/http://id.who.int/icd/entity/129180281 ]

  -There is a controversy among researchers over the overlap of depression and burnout the authors would be well advised in the Introduction to add a paragraph on this issue.

[ https://www.mdpi.com/2227-9032/10/1/134/htm    https://www.sciencedirect.com/science/article/abs/pii/S0272735815000173   ] . Also in the discussion it is useful to comment on their findings on this issue.

  1. Materials and Methods

- I would ask the authors to add a paragraph with data on the pressure that the pandemic exerted on the Polish health system during the research period.

  1. Results

 -I would suggest to the authors to transfer to appendix, tables of lesser importance such as table 1, as well as controls that are not statistically significant. In general, tables in the text should be made smaller to be more readable.

-I will ask the authors to add R squared change values to regression analysis (R-Squared is a statistical measure of fit that indicates how much variation of a dependent variable is explained by the independent variable (s) in a regression model).

Author Response

9th March 2022

Opole, Poland

Dear Editor

We would like to sincerely thank the Editorial Board and the Reviewer of your esteemed International Journal of Environmental Research and Public for their positive feedback and constructive recommendations improving our paper entitled: Predictors of occupational burnout of healthcare workers in Poland during the COVID-19 pandemic: a cross- sectional study (Manuscript ID: ijerph-1626225).  In this first round of revision, we focused our efforts strongly on the points made in your letter. We would like to respond to this opinion based on our careful revision, point by point, as you can see in the table below. Accordingly, the final version of the manuscript text includes the all necessary modifications and improvements.

Author's Reply to the Review Report:

REVIEWER #2

COMMENTS

AUTHORS’ REPLY

1.      Introduction. I would ask the authors to check again the definition proposed by the WHO for burnout, the WHO does not consider burnout a psychological syndrome but classifies it in the chapter "Problems associated with employment or unemployment". The definition given by WHO : «Burnout is a syndrome conceptualized as resulting from chronic workplace stress that has not been successfully managed. It is characterised by three dimensions: 1) feelings of energy depletion or exhaustion; 2) increased mental distance from one’s job, or feelings of negativism or cynicism related to one's job; and 3) a sense of ineffectiveness and lack of accomplishment. Burn-out refers specifically to phenomena in the occupational context and should not be applied to describe experiences in other areas of life» [  https://icd.who.int/browse11/l-m/en#/http://id.who.int/icd/entity/129180281 ]

The authors thank the reviewer for constructive recommendation improving the quality of our paper.

The definition of ‘burnout’ has been improved strictly according to reviewer’s comment.

2.      There is a controversy among researchers over the overlap of depression and burnout the authors would be well advised in the Introduction to add a paragraph on this issue.

The authors thank the reviewer for constructive recommendation. In the introduction it has been written:

‘The relationship between depression and burnout requires a separate discussion. The researchers' opinions are ambiguous on this point. Because of common etiological pathways and shared symptoms the singularity of the burnout phenomenon vis-à-vis depression is unclear [14]. Bianchi et al., climes that burnout may not be a separate psychological phenomenon but a dimension of depression [15]. In turn the findings of metanalysis conducted by Koutsimani et al revealed no conclusive overlap between burnout and depression and burnout and anxiety, indicating that they are different and robust constructs [16]. In the study of Pachi et al., the association between burnout and depression was confirmed. These authors believed that, despite the fact that a strong correlation coefficient was obtained between these variables, the interpretation of the variation of depression from burnout at the level of 43.7% cannot justify the overlap between burnout and depression. The authors adopted the position of the World Health Organization, which treats burnout as a separate disease [17]. We also adopted this concept in our study.’

3.      https://www.mdpi.com/2227-9032/10/1/134/htm    

https://www.sciencedirect.com/science/article/

abs/pii/S0272735815000173

Also in the discussion it is useful to comment on their findings on this issue

In the section discussion we wrote:’

‘The regression analysis conducted in our study showed that depression was a direct predictor of emotional exhaustion and job satisfaction and an indirect predictor of depersonalisation. The severity of depression adversely affected all three components of job burnout. There is the widespread but controversial opinion that burnout is a prodromal syndrome of depression [46]. The study conducted among nurses by Pachi et al also highlighted that burnout is a factor influencing depression. In this study regression evidenced that 43.7% of the variation in the BDI (Beck’s Depression Inventory) rating was explained by the CBI (Copenhagen Burnout Inventory) [17]. Kaschka et al also indicated that burnout is likely to be a risk factor for developing depression [47]. As we indicated in introduction there is a lot of controversy among researchers over the overlap of depression and burnout. It is certainly advisable to conduct further research in this area.’

4.      Materials and methods. I would ask the authors to add a paragraph with data on the pressure that the pandemic exerted on the Polish health system during the research period.

In section material and methods it has been written:” Health at a Glance data shows, that before Covid – 19 pandemic countries in Europe, such as Poland, Latvia and Romania, had fewer doctors and nurses per population comparatively to Germany, Norway or Sweden etc. Therefore, Poland had less capacity to respond to the pandemic. In Poland the number of practising doctors per 1 000 population equals 2.4 and practising nurses – 5.1 respectively (the average for EU countries is 3.8 for practising doctors and 8.2 for practising nurses).  Insufficient human resources made it difficult for healthcare professionals in Poland to respond to the sharp increase in demand for care [28].  Such a situation could initiate or aggravate the burnout syndrome in these people.

Sagan et al. climes that Poland have been overly reliant on their relatively high bed capacity, but this could not be supported with sufficient health workforce capacity. During the period of our research, in Poland not developed effective find, test, trace, isolate and support systems over the summer despite having relaxed most of the transmission protection measures since late spring. This left Poland ill-prepared for the rise in the number of COVID-19 infections they have been experiencing [29].’

5.      Results. I would suggest to the authors to transfer to appendix, tables of lesser importance such as table 1, as well as controls that are not statistically significant. In general, tables in the text should be made smaller to be more readable.

We have refined all the tables in terms of graphics. We placed tables 1,4,5,6  horizontally (previously they were vertically). We belived that at this moment the tables will be more readable. We believe that statistically insignificant data are also important for the reader.

6.      Results. I will ask the authors to add R squared change values to regression analysis (R-Squared is a statistical measure of fit that indicates how much variation of a dependent variable is explained by the independent variable (s) in a regression model).

In the section “results” it has been written:

‘The R² coefficient for the emotional exhaustion model was 0.534, which means that this model explains 53.4% of the variability on the emotional exhaustion scale. However, the remaining 46.6% depends on variables not included in the model and on random factors.’

‘The R² coefficient for the depersonalization model was 0.286, which means that this model explains 28.6% of the variation in the depersonalization scale. The remaining 71.4% depends on variables not included in the model and on random factors. ‘

‘The R² coefficient for the job satisfaction model was 0.358, which means that this model explains 35.8% of the variation in the job satisfaction scale. The remaining 64.2%% depends on the variables not included in the model and on random factors. ‘

We very much hope that our revisions appear comprehensive and proves helpful in obtaining a positive final decision accepting our paper for publication in your prestigeous journal. Awaiting your decision, I remain with kind regards

Authors

Reviewer 3 Report

This paper reports new data for a Polish sample of health care workers.   My comments on it are as follows:

A major concern with this study is the mismatch between the first stated research question (“…the analysis of the burnout phenomenon, the level of anxiety and depression and the quality of life among healthcare workers in the times of COVID-19 pandemic…”) and the method chosen to address it.  The question is attempting to make population prevalence estimates - the numbers of anxious, depressed and burned-out staff in the health workforce – but is attempting to do this via ‘non-probability sampling’ using an online web questionnaire and snowballing.  We are told 1508 people got as far as opening the CAWI web page but not what proportion of the targeted workforce this was.  We are then told that of those who opened the CAWI page 1008 failed to complete the survey and only 497 (33%) provided complete data.  We thus we probably have a very small percentage of the workforce surveyed and it is quite likely that there are serious response biases here.  Are the burned-out/depressed/anxious more likely to want to express their unhappiness than others?  Are some people simply too busy and over-worked to want to complete the survey?  Why did 66% start the survey but not finish it? What prevented completion and are those reasons systematically related to the key variables being studied? The authors note that more females than males responded. While these questions remain unanswered it is not possible to interpret the prevalence rates reported in a meaningful way and it is telling that the figures presented here are higher than in other studies (line 370 etc.).  I have no doubt that COVID has led to some serious burnout but without data on pre-COVID levels and these issues of potential sampling bias being addressed we don’t really know the size of the effect of COVID. 

I recognise that the authors are aware of these problems and have noted that in order to do the study quickly some compromises have had to be made.   Despite this, quite strong conclusions are drawn at the end of the paper and really these need to be toned down.  Some form of weighted analysis using the known characteristics of the Polish health care work force would be a good idea.

A second concern is with the potential conceptual overlap between the measures of aspects of  burnout, depression, anxiety and QoL.  If I am depressed, for example, I am unlikely to rate my quality of life as high because my levels of depression are part of my quality of life.  Similarly, some aspects of burnout are also symptoms of anxiety and depression.  Given the vague definitions of these latent constructs it is important to establish their conceptual distinctiveness. If they are not conceptually distinct then the attempts to predict the facets of burnout from other variables is potentially to engage in a tautological exercise.  What would be good to see is a confirmatory factor analysis at the item level to establish the distinctiveness of the measures of the key constructs.

Minor points and typos:

Line 54 – ‘exposition’ should be ‘exposure’?

Line 84 – ‘a greater number of overtime’ doesn’t make sense. Do you mean a greater number of overtime shifts?

Table 4 – as the psychological variables are; a) latent and b) measured on wildly different scales where the meaning of the scale intervals is unknown it would be more informative to report standardised regression coefficients.

A lot of inferential tests have done with no protection against likely Type 1 error rates – some form of protection (Bonferroni or Bonferroni-Holm corrections for e.g.) might be appropriate.

Author Response

Dear Editor

We would like to sincerely thank the Editorial Board and the Reviewer of your esteemed International Journal of Environmental Research and Public for their positive feedback and constructive recommendations improving our paper entitled: Predictors of occupational burnout of healthcare workers in Poland during the COVID-19 pandemic: a cross- sectional study (Manuscript ID: ijerph-1626225).  In this first round of revision, we focused our efforts strongly on the points made in your letter. We would like to respond to this opinion based on our careful revision, point by point, as you can see in the table below. Accordingly, the final version of the manuscript text includes the all necessary modifications and improvements.

Author's Reply to the Review Report:

REVIEWER #3

COMMENTS

AUTHORS’ REPLY

1.      A major concern with this study is the mismatch between the first stated research question (“…the analysis of the burnout phenomenon, the level of anxiety and depression and the quality of life among healthcare workers in the times of COVID-19 pandemic…”) and the method chosen to address it.  The question is attempting to make population prevalence estimates - the numbers of anxious, depressed and burned-out staff in the health workforce – but is attempting to do this via ‘non-probability sampling’ using an online web questionnaire and snowballing.  We are told 1508 people got as far as opening the CAWI web page but not what proportion of the targeted workforce this was.  We are then told that of those who opened the CAWI page 1008 failed to complete the survey and only 497 (33%) provided complete data.  We thus we probably have a very small percentage of the workforce surveyed and it is quite likely that there are serious response biases here.  Are the burned-out/depressed/anxious more likely to want to express their unhappiness than others?  Are some people simply too busy and over-worked to want to complete the survey?  Why did 66% start the survey but not finish it? What prevented completion and are those reasons systematically related to the key variables being studied? The authors note that more females than males responded. While these questions remain unanswered it is not possible to interpret the prevalence rates reported in a meaningful way and it is telling that the figures presented here are higher than in other studies (line 370 etc.).  I have no doubt that COVID has led to some serious burnout but without data on pre-COVID levels and these issues of potential sampling bias being addressed we don’t really know the size of the effect of COVID. 

In the section materials and methods we wrote:"

‘The available data show that 234,117 nurses, 28,899 midwives [30], 145,659 doctors [31], 26,495 physioterapists are employed in Poland [32]. The Central Statistical Office in Poland reports that 11,000 paramedics work in medical rescue teams [33]. With a confidence level of 95%, a margin of error of 5%, p = 50% the minimum study sample was set at 1896 subjects. Initially, 1508 respondents got interested in the study (they opened the electronic questionnaire). The group of 1008 participants started filling in the form but did not complete it. The questionnaire required the respondents to answer all the questions compulsorily otherwise it could not be sent to the database. Therefore, eventually, 500 people (33.3%) completed all the questionnaires properly. Three of them appeared to be medical field students and their questionnaires had to be excluded from the study. Finally, the research was based on 497 well-completed questionnaires (maximum error 4%, confidence level of 95%, p=50%).’

Moreover, in the section ‘limitations of the study’ we wrote:

'We collected data from 497 subjects. We are not sure why the rest of the survey participants did not complete the survey, even though they initially expressed interest in the topic of our research. We can only assume that the difficult time of the pandemic and excessive workload of healthcare workers could discourage the respondents from filling in a fairly extensive package of questions (the questionnaires contained a total of 80 questions). However, in order to thoroughly research the phenomenon of burnout from the perspective we are interested in, we decided that the tools we chose are necessary in the study '

The research by Gualamo et al. [39] actually showed a much lower percentage of people with burnout, but the next studies show results similar to the results of their own research - Luceno-Moreno et al. [24], Barello et al. [23], Khasne and co-workers [40], Jacome et al. [41], Pniak et al. [42]. In our own research, unfortunately, we do not have data on occupational burnout among healthcare workers in Poland before the pandemic, so we actually do not know the effect of the COVID-19 pandemic on the level of occupational burnout in the surveyed group. In the discussion, we have added excerpts about the level of professional burnout of healthcare workers in Poland before the pandemic. We agree that one should be careful in drawing strong conclusions. We partially revised our conclusions.

2.      I recognise that the authors are aware of these problems and have noted that in order to do the study quickly some compromises have had to be made.   Despite this, quite strong conclusions are drawn at the end of the paper and really these need to be toned down.  Some form of weighted analysis using the known characteristics of the Polish health care work force would be a good idea.

Due to the small sample of respondents and information from the pre-pandemic burnout literature, we have toned down our conclusions and added information about the sample size to the limitations of the study. Information on the lack of own research on burnout before the COVID-19 pandemic has already been included in the limitations of the study.

3.      A second concern is with the potential conceptual overlap between the measures of aspects of  burnout, depression, anxiety and QoL.  If I am depressed, for example, I am unlikely to rate my quality of life as high because my levels of depression are part of my quality of life.  Similarly, some aspects of burnout are also symptoms of anxiety and depression.  Given the vague definitions of these latent constructs it is important to establish their conceptual distinctiveness. If they are not conceptually distinct then the attempts to predict the facets of burnout from other variables is potentially to engage in a tautological exercise.  What would be good to see is a confirmatory factor analysis at the item level to establish the distinctiveness of the measures of the key constructs.

Burnout, anxiety, depression and QoL have been scientifically researched with tools that have been previously validated. These tools have undergone standardization and normalization processes. Moreover, they are accurate and reliable. Relevance means that a given tool measures exactly what its creators intended to measure.

  The confirmatory factor analysis was part of the validation of the above-mentioned tools. So there is no need to do it again.

4.      Line 54 – ‘exposition’ should be ‘exposure’?

Line 54 has been improved.

5.      Line 84 – ‘a greater number of overtime’ doesn’t make sense. Do you mean a greater number of overtime shifts?

Line 84 has been improved.

6.      Table 4 – as the psychological variables are; a) latent and b) measured on wildly different scales where the meaning of the scale intervals is unknown it would be more informative to report standardised regression coefficients.

In the models, we have not only psychological variables, but also variables that are not expressed in numbers, such as gender, education, etc. For such variables, the interpretation of standardized coefficients is very troublesome. Therefore, it was decided to use not standardized coefficients.

We consulted the reviewer's remark with a professional statistician who performs statistical analyzes in the field of medicine and health sciences. This statistic has very good references from many scientists.

https://lukaszderylo.pl/referencje.html

7.      A lot of inferential tests have done with no protection against likely Type 1 error rates – some form of protection (Bonferroni or Bonferroni-Holm corrections for e.g.) might be appropriate.

The lack of this protection is by design. The idea is to be able to compare the results of univariate and multivariate analyzes and on this basis determine which variables are direct predictors of burnout. This would not have been possible after the proposed corrections had been applied.

We very much hope that our revisions appear comprehensive and proves helpful in obtaining a positive final decision accepting our paper for publication in your prestigeous journal. Awaiting your decision, I remain with kind regards

Authors

Round 2

Reviewer 3 Report

The authors have toned down their claims and have included a description of the pre-COVID Polish workforce as suggested.

My original review focussed on two issues; 1) possible response biases arising out of the method of sampling and lack of data to allow the study of non-responders and 2) potential conceptual overlap in some of the measures used.   The authors are arguing that it is not necessary to address these other than to tone down the claims.  It is for the editors to decide whether this is sufficient to make the paper worthy of publishing. 

I think the suggestion that because the psychometric measures of burnout/depression/anxiety etc. have been validated elsewhere is inviting the reader to make a leap of faith that these measures work as intended in the present sample of potentially stressed Polish health care workers.  The reliability of a test, for example, is an index of its reliability in a specific sample, it is not a universal characteristic of a test that can be carried across samples.  The authors report reliabilities from other Polish studies but how reliable are the measure in the present study? Similarly, construct validity as established via factor analysis is also sample-specific and one of the many problems in psychology is the readiness to accept the validity, and particularly discriminant validity, of measures without checking to confirm that these assumption are justified.

I accept the authors’ point about the utility of standardised coefficients when referring to binary predictors like gender but for the other predictors the unstandardised coefficients remain difficult to interpret and the relative importance of these isn’t clear.

Author Response

Dear Editor

We would like to sincerely thank the Editorial Board and the Reviewer of your esteemed International Journal of Environmental Research and Public for their positive feedback and constructive recommendations improving our paper entitled: Predictors of occupational burnout of healthcare workers in Poland during the COVID-19 pandemic: a cross- sectional study (Manuscript ID: ijerph-1626225).  In this first round of revision, we focused our efforts strongly on the points made in your letter. We would like to respond to this opinion based on our careful revision, point by point, as you can see in the table below. Accordingly, the final version of the manuscript text includes the all necessary modifications and improvements.
